# Children with disabilities in nutrition programmes: Thematic analysis of training and guidelines

Katie Fulford[1,2]*, Emily DeLacey[2,3], Fiona Beckerlegge[4], Julia Hayes[3], Himali de Silva[5], Marko Kerac[1,2]

**1** Department of Population Health, Faculty of Epidemiology and Population Health, London School of Hygiene & Tropical Medicine, University of London, London, United Kingdom, **2** Centre for Maternal, Adolescent, Reproductive & Child Health (MARCH), London School of Hygiene & Tropical Medicine, University of London, London, United Kingdom, **3** Nutrition and Health Services, Holt International, Eugene, Oregon, United States of America, **4** Kyaninga Child Development Centre, Fort Portal, Uganda **5** Evelina London Children's Community Speech and Language Therapy Service, Mary Sheridan Centre for Child Health, London, United Kingdom

* lsh2300281@alumni.lshtm.ac.uk

## Abstract

Children with disabilities are often excluded in nutrition policy, programming and practice. A recently established "Feeding and Disability Resource Bank" collates currently available guidelines and training materials for children with disabilities. Our research aims to evaluate these resources to improve quality of care for children with disabilities in nutrition programming and services. Objectives include analysis of resources, evaluating gaps in resources, and barriers and facilitators to their use. Methods include thematic analysis and semi-structured key-informant interviews to identify priority areas for improvement of the resources and provide context to their use. Data were analysed deductively to identify themes around the study's objectives. Analysis of the Feeding and Disability Resource Bank included 13 training resources, eight programme packages and nine programmatic guidelines which were mapped for coverage and depth of recommendations for 28 key topics. Twelve key informant interviews were conducted. Synthesized key topics were identified by key informants and thematic analysis included identification of disability, feeding difficulties and malnutrition, management of feeding difficulties and malnutrition and integration of care for a child, family and community. Findings indicate there are some high-quality resources to support children with disabilities, but more are needed to provide comprehensive care. Gaps include appropriate assessment of nutritional status (8/30 resources), micronutrient deficiencies (0/30), management of the nutritional quality of a modified diet (6/30) and dehydration (3/30). Resources can be strengthened by filling identified gaps, standardising recommendations and operationalising how these practices can be integrated into programming. The uptake of these recommendations could be maximized by concurrent consideration of the visibility of disability in

**Data availability statement:** Anonymised transcripts from the interviews are available on the LSHTM Data Compass Repository (DOI: https://doi.org/10.17037/DATA.00004807).

**Funding:** The authors received no specific funding for this work.

**Competing interests:** I have read the journal's policy and the authors of this manuscript have the following competing interests: Emily DeLacey and Julia Hayes work for Holt International and authored one of the guidelines in this review. Himali de Silva has volunteered for health charity MAITS, whose materials were also reviewed.

nutrition agendas, accessibility and dissemination of the resources, and the human and resource capacity of the sector. Unless addressed, children with disabilities will continue to experience systemic exclusion from accessing quality care, treatment and prevention of further disability.

## Introduction

Malnutrition continues to be a significant threat to child health and development globally. UNICEF/WHO/World Bank statistics estimate that some 148 million children younger than 5 years worldwide (22.3%) are stunted, 45 million (6.8%) wasted and 37 million (5.6%) overweight. [1] Millions are also vulnerable to micronutrient deficiencies such as iron deficiency anaemia. [2] Fewer than half of all countries are "on track" to meet even one of the nine global nutrition targets by 2025. [3] The rate of current progress will result in the 2030 malnutrition targets being missed by more than 39 million children. [1] Some of those most vulnerable are the 240 million children with disabilities, about one in every 10 children. [4–6] A child's likelihood of having a disability by their fifth birthday is now 10 times higher than their likelihood of dying. [7] This reflects both successful child mortality interventions, as well as the neglect of morbidity and prevention in those interventions. Between 1990 and 2016 Sub-Saharan Africa saw a 55% reduction in mortality for children younger than 5 years, and a 71.3% increase in developmental disorders (such as epilepsy, neurodevelopment disability, and autism spectrum disorder). [8]

Malnutrition is well recognized as a major cause of disability at all stages of the life course, particularly during the critical window of the first 1,000 days of life. [9,10] Maternal and neonatal disorders, where nutrition plays a crucial role, are the leading cause of all disability-adjusted life years (DALYs) (52.47%) and nutritional deficiencies are the third (15.26%). [11] The reverse is also true — children with a disability are at greater risk of malnutrition and consequential ill health. [12] A pooled meta-analysis of data from LMICs shows that children with disabilities had almost three times the odds of being underweight, twice the odds of wasting and twice the odds of stunting when compared to children without. [13,14] Many factors influence the nutritional status of children with disabilities including presence of feeding difficulties, access to quality care, age-appropriate nutrient-rich foods, feeding practices, environmental factors, resources and societal/cultural norms. [15] Difficulties with feeding can be an early indicator of delayed development or disability risk later in life. Timely identification of these provides practitioners with a golden opportunity to better support families both short- and long-term.

With malnutrition and disability both being major factors in global health, and clear evidence demonstrating their bi-directional risk relationship, stronger links could produce mutual benefits and improved outcomes for children. [16] The Convention of the Rights of Persons with a Disability and the Convention of the Rights of the Child mandate all children have the right to accessible and affordable treatment for any disease, illness or health condition. [17,18]

Many current nutrition programmes fail to adequately consider or support underlying disability. [19,20] A recent scoping review identifies insufficient policies, programmes or evidence to support children with disabilities or feeding difficulties in malnutrition prevention and treatment. [21] Resources in this review are part of a publicly available bank. [22,23] To optimize future use and value, it is important to understand the details of what these resources and guidance documents currently cover. This study aims to evaluate the resources and guidelines available on this resource bank, to help professionals supporting children with disability better identify the opportunities for disability-inclusive programming, prioritise areas for development and strengthening of policy, programming and research, ultimately to help improve quality of care for children with disabilities in nutrition programming.

To achieve this, our objectives were to:

1. Conduct a thematic analysis of the resources available in the Feeding and Disability Feeding Bank. [22]

2. Evaluate whether these resources adequately address the main challenges and needs in nutrition programming to support children with disabilities.

3. Evaluate the barriers and facilitators to use of these resources.

Together the results of this analysis can be used to generate recommendations for future resource development, nutrition agenda and policy guidance.

## Materials and methods

### Study design

This is a mixed methods study using cross sectional data and qualitative key-informant (KI) interviews to explore the currently available resources for supporting children with disabilities in nutrition programmes. We follow the STROBE checklist with secondary additions from the COREQ checklist for qualitative research.

### Ethics statement

This study was conducted in accordance with the Declaration of Helsinki and approved by the London School of Hygiene and Tropical Medicine's Ethics Committee (ref: 30069). Informed consent was obtained from all participants involved in the study, including written consent to publish this paper.

### Feeding and disability bank review

We extracted data from a recently compiled global bank of resources intended as a repository of materials to help nutrition and disability programme managers, government leaders and other stakeholders design and implement effective nutrition programmes for children with disabilities. [21,22] The bank is now facilitated by UNICEF, although only resources present on the original USAID Feeding and Disability Resource Bank were included in this review. [22,23]

For inclusion in our study, resources needed to be published in English, aimed toward professional audiences, open access and accessible during the search period of 1st June – 31st July, 2024. They also needed to include references to both nutrition and disability or feeding difficulties, with any disability or feeding difficulty considered. [21–23] Resources focused on professional development of the healthcare worker were excluded from our analysis. These decisions were made to focus on what resources and guidelines are available to professionals for development of skills related to managing disability and malnutrition, establishing or implementing programmes, and the implications for policy. Individual screening/assessment tools on the resource bank were excluded from the main thematic analysis as it was outside the scope of this analysis to assess the efficacy of any individual tool, but are discussed separately in Table 3.

Global Public Health

PLOS

**Key-Informant interviews**

Key-informant (KI) interviews were conducted to complement and provide a contextual backdrop to the results and implications of the thematic analysis. A phenomenological approach was adopted to explore the real world experience of accessing and using these resources. A mixture of purposive and snowball sampling was used to identify and invite professionals to share their expertise in a KI interview. KIs were targeted based on their experience in global nutrition and child disability at either a programme or policy level, including those with direct service experience. KIs were from a variety of different professional backgrounds covering medicine, nutrition/dietetics, child development and other allied-health professions, at varying stages of their careers, working across diverse sectors and covered a wide variety of geographies (Table 1). Purposive sampling began with four participants, with remaining KIs snowball sampled between 24th June – 16th July 2024.

Information was provided and written consent obtained prior to interviews commencing. The interviews were conducted via Zoom between 9th July - 7th August, 2024. The interviews were conducted with participants in 8 different countries, with collective professional experience working across Asia and the Pacific, Middle East and North Africa, Sub-Saharan Africa, Europe and North America. Meetings were recorded and transcribed using Zoom's built-in audio transcription software.

**Table 1. Description of Key Informants.**

| Description of 12 Key Informants | Number % (percentage) |
|---|---|
| **Sex** | |
| Female | 11 (92) |
| Male | 1 (8) |
| **Professional Background** | |
| Medicine | 2 (17) |
| Nutrition/Dietetics | 5 (41) |
| Programme Specialist/Child Development | 2 (17) |
| Allied-health sectors (Speech and language and Physiotherapy) | 3 (25) |
| **Professional Sector** | |
| Health Sector | 3 (25) |
| Non-Governmental Organisation | 6 (50) |
| Governmental Organisation | 1 (8) |
| UN or Other International Agency | 2 (17) |
| **Years of Experience Post-Qualification** | |
| <10 | 3 (25) |
| 10-20 | 3 (25) |
| >20 | 6 (50) |
| **Geographical Location** | |
| Sub-Saharan Africa | 3 (25) |
| Europe | 3 (25) |
| North America | 6 (50) |
| **Number of Participants with Experience in Each Global Region** | |
| Asia and the Pacific | 7 (58) |
| Middle East and North Africa | 2 (17) |
| Sub-Saharan Africa | 10 (83) |
| Europe | 5 (41) |
| North America | 7 (58) |

The transcriptions were then checked and edited in Microsoft Word by the interviewer, using the audio recording of the meeting. Transcripts were made anonymous through the removal of any identifiable information and categorisation of job roles, projects, locations and organisations.

KI interviews were semi-structured following a topic guide aligned with study objectives (S1 File). The topic guide started with questions on introduction and background of the participant. Then there were three main sections, following the study objectives; Strengths and Weaknesses of the Resources, Barriers and Opportunities to Implementation, and Future Development of the Resources. There were 11 main questions with zero to four probing questions for each. The topic guide was pilot tested for feasibility with practice interviews. Modifications of this guide were agreed upon by authors.

### Data collection and coding

The data collection and analysis were conducted from the UK. Included resources were categorised as programme packages, programmatic guidance or training manuals. All were reviewed in detail, and topics covered were coded to identify key content areas. Finally, these were grouped into themes. A heat map demonstrated which of the 28 key content areas were identified in each resource and whether any programmatic recommendations were given. This coverage and quality of recommendations followed a key to reduce reviewer bias.

The KI interviews were coded and analysed following a three-step practical thematic analysis approach for multi-disciplinary health research, including reading, coding and theming. [24] Coding was done to identify arising strengths and weaknesses of the resources as well as external facilitators and barriers to their implementation. Codes were then grouped into themes within their brackets of strengths, weaknesses, facilitators and barriers.

### Analysis

Interview data on the strengths and weaknesses of various resources were matched and discussed together with the 28 key areas from the thematic analysis: identification, management and integration of care. This combination of results highlights what the resources and guidance currently cover, and how they meet the needs of professionals supporting children with disability. Presenting key topics in heat maps helps professionals seeking information on a particular topic to easily and rapidly identify the best resources.

Interview data on the barriers and facilitators to the resource's implementation were analysed and reported separately. This enables identification of opportunities for greater disability-inclusive programming, priority areas for development and strengthening of policy, programming and research.

## Results

### Feeding and disability bank review

Of the 86 resources on the Feeding and Disability Resource Bank, 30 met the inclusion criteria for thematic analysis: thirteen training resources, eight programme packages and nine programmatic guidelines (Fig 1). The resources were published in 11 different countries in North America, Europe, East and South-East Asia, and Africa and collectively include versions in 36 different languages.

There were 28 key content areas identified within the literature, under three main themes; identification, management and integration of care. The first, identification, focused on five key content areas on identification of children with disabilities, feeding difficulties and/or malnutrition. The second included 15 content areas related to managing feeding difficulties and malnutrition, such as recommendations for positioning, feeding equipment, oral support and wider cognitive and socio-emotional development. Finally, the third content area consisted of eight topics focused on integration of disability services into wider care including follow-up and holistic care for children, support for caregivers and addressing environmental barriers such as exclusion, accessibility and stigma.

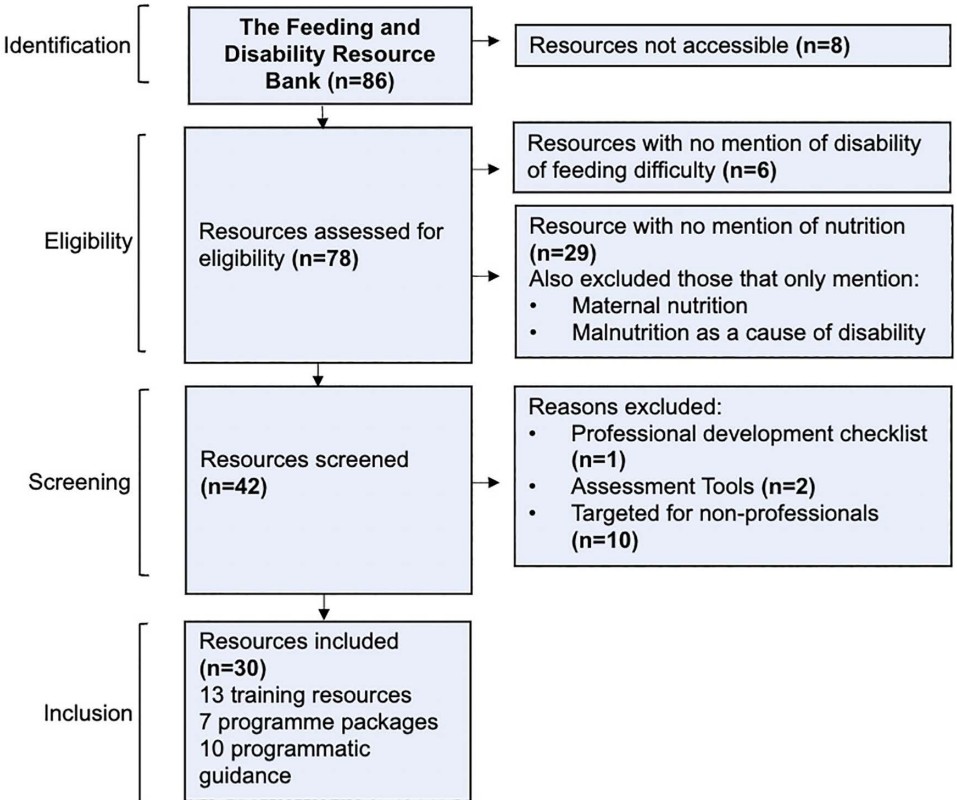

**Fig 1. Flow Diagram depicting inclusion and exclusion criteria of resources from the Feeding and Disability Resource Bank for thematic analysis.**

## Key informant interviews

Of the 14 professionals invited for participation, one KI did not respond and one declined due to a perceived lack of specific knowledge on the subject (Table 1). Twelve interviews were conducted Table 2.

## Disability identification

Identifying disability is paramount to the appropriate management of nutrition and health. Developmental monitoring is one method utilised to screen children for early signs of disability. Of the 30 resources, 12(40%) reference a standardized assessment tool for monitoring developmental and disability (Table 3). These are mostly in programmatic guidance, as only two are training manuals and two programme packages. Training manuals and programme packages more commonly contain information on the developmental milestones (40%), which equip health workers to clinically assess evidence of age-appropriate development. Guidance on how development milestones relate to feeding skills, such as use of the tongue to move food to the back of the mouth for swallowing or ability to hold feeding utensils, was only found in five of the resources (17%). In low-resource settings, standardized assessment tools can aid less-experienced or semi-skilled health workers to appropriately identify children with disability or developmental delay and manage accordingly, particularly due to the heterogeneity of how different disabilities present and children's individual functional abilities.

**Table 2. Heat map depicts coverage and recommendations of key content areas under the themes of identification of disability, feeding difficulties and malnutrition, within the resources from the Feeding and Disability Resource Bank [22]. Red indicates not acknowledged; light blue: acknowledged but no guidance; medium blue: acknowledged and some guidance (1-2 recommendations); and dark blue: acknowledged and comprehensive guidance (>3 recommendations).**

| Type | Resource Name | Publisher | Year | Assessment of Disability | Developmental Milestones | Assessment of Feeding | Clinical Signs of Feeding Difficulty | Assessment of Nutritional Status |
|---|---|---|---|---|---|---|---|---|
| TM | Feeding in Children with Cerebral Palsy [25] | Physiopedia | 2022 | medium blue | red | red | dark blue | red |
| | Resource Library [26] | SPOON | 2022 | red | red | medium blue | medium blue | medium blue |
| | Introduction to Nutrition and Feeding for Children [27] | SPOON | 2021 | medium blue | light blue | red | red | red |
| | Nutrition integration for Children with Special Needs [28] | LEARN | 2020 | medium blue | red | red | dark blue | medium blue |
| | Feeding and Positioning Manual: Guidelines for Working with Babies and Children [29] | Holt International | 2019 | red | dark blue | red | dark blue | red |
| | Working with Infants with Feeding Difficulties [30] | MAITS | 2018 | red | dark blue | medium blue | dark blue | red |
| | Working with Children with Eating and Drinking Difficulties [31] | MAITS | 2017 | red | dark blue | dark blue | dark blue | light blue |
| | Our Videos: Care of Small Babies [32] | GHM | 2017 | red | red | red | medium blue | light blue |
| | How to Approach Feeding Difficulties in young children [33] | H. R. Yang | 2017 | red | red | red | medium blue | medium blue |
| | Assessment of Neurodisability and Malnutrition in Children in Africa [34] | Gladstone | 2014 | dark blue | red | light blue | light blue | medium blue |
| | Growth Charts for Children with Special Health Care Needs [35] | CDC | 2014 | red | red | red | red | dark blue |
| | Assessment of Growth and Nutrition in Children with Cerebral Palsy [36] | Samson-Fang | 2013 | red | red | red | red | medium blue |
| | Feeding Problems in Infancy and Early Childhood [37] | Arts-Rodas | 1998 | dark blue | red | medium blue | medium blue | light blue |
| PP | Responsive Care and Early Learning Addendum [38] | USAID | 2022 | red | light blue | red | dark blue | red |
| | Child Nutrition Program: Community Flipbook [39] | Holt International | 2022 | red | red | red | red | red |
| | Ubuntu Manual [40] | Ubuntu | 2021 | red | medium blue | red | dark blue | red |
| | Baby Ubuntu [41] | Ubuntu | 2021 | red | medium blue | red | dark blue | red |
| | Juntos [42] | Ubuntu | 2019 | red | medium blue | medium blue | dark blue | light blue |
| | Working with Children with Developmental Disabilities and Caregivers [43] | MAITS | 2018 | medium blue | medium blue | medium blue | dark blue | light blue |
| | Supporting Families for Nurturing Care [44] | UNICEF | 2016 | dark blue | medium blue | medium blue | red | red |
| | Prevention Toolkit on Cerebral Palsy [45] | CBM | 2012 | red | light blue | light blue | light blue | red |

*(Continued)*

**Table 2.** (Continued)

| Type | Resource Name | Publisher | Year | Assessment of Disability | Developmental Milestones | Assessment of Feeding | Clinical Signs of Feeding Difficulty | Assessment of Nutritional Status |
|---|---|---|---|---|---|---|---|---|
| PG | Seen, Counted, Included [5] | UNICEF | 2022 | (dark blue) | (red) | (red) | (red) | (red) |
| | Essential Actions on Disability Inclusive Nutrition [46] | UNICEF | 2022 | (dark blue) | (red) | (red) | (red) | (light blue) |
| | Inclusion of Children and Young people with Disabilities in Routine General Healthcare [47] | UNICEF | 2022 | (dark blue) | (red) | (red) | (red) | (light blue) |
| | Improving Young Children's Diets during Complementary Feeding [48] | UNICEF | 2020 | (red) | (red) | (red) | (red) | (red) |
| | Disability Inclusive Health Services Toolkit [49] | WHO | 2020 | (red) | (red) | (red) | (red) | (red) |
| | Monitoring Children's Development in Primary Care Services [50] | WHO | 2020 | (dark blue) | (pale blue) | (red) | (red) | (red) |
| | Nurturing Care Handbook, Strategic Action 4 [51] | WHO | 2021 | (dark blue) | (red) | (red) | (red) | (red) |
| | Including Children with Disabilities in Humanitarian Action: Nutrition [52] | UNICEF | 2018 | (dark blue) | (red) | (red) | (light blue) | (light blue) |
| | A Toolkit for Measuring Early Child Development in LMICs [53] | WB | 2017 | (dark blue) | (red) | (red) | (red) | (red) |

** TM: Training Manual, PP: Programme Package, PG: Programmatic Guidance

**Table 3. Developmental screening and feeding assessment tools in the feeding and disability resource bank.**

| | Assessment Tool | Publisher | Year | Development | Disability | Feeding |
|---|---|---|---|---|---|---|
| 1 | Malawi Development Assessment Tool: IDEC Monitoring Tool Version 1.3 [56] | University of Liverpool | 2020 | x | | |
| 2 | Washington Group Short Set on Functioning (WG-SS) [54] | Washington Group- on Disability Statistics | 2020 | x | | |
| 3 | Module on Child Functioning Questionnaires [55] | UNICEF and Washington Group | 2016 | x | | |
| 4 | Modified Checklist for Autism in Toddlers, revised, with follow-up (M-CHAT-RF) [57] | D. Robins | 2018 | | x | |
| 5 | Infant and Child Feeding Questionnaire [58] | Feeding Matters | 2020 | | | x |
| 6 | Eating and Drinking Ability Classification System (EDACS) [59] | Sussex Community NHS Foundation Trust | 2013 | | | x |
| 7 | International Dysphagia Diet Standardisation Initiative Framework [60] | IDDSI | 2019 | | | x |

*"We were just having these re-admissions, and it meant in terms of disability they were getting further and further behind their developmental milestones because we were not addressing the root cause of the malnutrition [… that the child had a disability]." Participant 3*

*"We're relying on professional knowledge and skills and clinical reasoning to be able to adapt accordingly. That's what makes it very difficult when you've lower-level health workers or semi-skilled health workers who don't have that." Participant 2*

The most referenced assessment tools are the Washington Group Short Set on Functioning and UNICEF's Module on Child Functioning Questionnaires [54,55]. Few recommendations for developmental monitoring are specific to the context of nutrition programming. Resources, such as the World Bank Toolkit for Measuring Early Child Development in Low- and Middle-Income Countries and Monitoring Children's Development in Primary Care [50,53], indicate nutrition services as an opportunity for developmental monitoring but give no specific recommendations how to integrate the practice into that context. The Washington Group Questions are for children aged 5 years and older, and the Child Functioning Module for 2–4 years and 5–17 years [54,55]. In the context of a nutrition programme, these assessment tools are relevant only outside of the first 1,000 days of life, where interventions to support development, nutrition and health are known to have the greatest effect and long-term impact.

> *"Under 6 months it's difficult to diagnose [developmental delay and disability] as they're not going to fit the screening tools. This is why there is this focus on feeding difficulties rather than outright disability. Moving it so it is less stigmatized, so if the mother is saying 'my infant takes a long time to feed', there are things we could do then." Participant 4*

**Identification of feeding difficulties**

Only 5/30 (17%) resources describe how to conduct a feeding assessment and none refer to the standardized feeding assessment tools currently accessible in the resource bank: the Eating and Drinking Classification System for Children with Cerebral Palsy (EDACs) and the Infant and Child Feeding Questionnaire (Table 3). The resource bank also contains a classification system of modified diets for the management of dysphagia, the IDDSI Framework, which is similarly not referenced in any of the other resources but currently a standard tool used in the field of nutrition. Guidance on identifying clinical signs of feeding difficulties are included in nine of the training resources (70%) and five programmes (63%).

**Monitoring nutritional status and growth**

Recommendations for monitoring nutritional status and growth, specific to children with disability, are only described in 8/30 resources (27%). None of the programme packages give recommendations on identifying malnutrition or measuring nutritional status.

> *"Classically malnutrition is still [identified] through anthropometry, and you can't measure weight and height on a lot of these children, depending on what the disability is. MUAC will be good for some, but not for others." Participant 4*

As MUAC (Mid-Upper Arm Circumference), weight and height are standard global measurement indicators, many guidelines such as the WHO 2023 Wasting Guidelines or Sphere standards use them for screening, monitoring and eligibility criteria for treatment. [61,62] Consequently, many children with disabilities are excluded from appropriate assessment and therefore stopped from receiving treatment and support.

> *"A lot of children won't be picked up by malnutrition programmes because they say 'they have a disability, there's no guidelines to screen these children and we don't know how to include them, track their progress, determine if they're improving or not improving.'" Participant 5*

The CDC Growth Charts for Children with Special Healthcare Needs contains the most comprehensive guidance for adapting anthropometry practices. [35] It describes how the most important practices in growth monitoring are not limited to which measurement is taken but the consistency, accuracy in technique and repetition of measurement over time. The guidance includes alternative measurement options such as crown-to-rump length recumbent or sitting, arm-span or

segmental lengths such as upper-arm and lower-leg length which can also be plotted on growth charts to monitor a pattern. Tracking indicators that can be measured, and having a formal approach to recording those measurements, provides a reference point to monitor an individual child. Wider guidance could reiterate this and provide caveats for how to identify, admit and monitor children in programmes when anthropometry is not possible.

> *"Even though there's no internationally agreed global guidance on it, you can at least track children month to month, and they should not be worsening from where they started. If they can't extend their legs, we're not taking a length measurement, but will still get weight, head circumference, MUAC." Participant 5*

Recommendations in the training manuals and guidelines were only for anthropometry and growth monitoring; there was no mention of measuring micronutrient deficiencies or clinical signs of malnutrition.

**Management**

Table 4 shows recommendations covering the different aspects of management are varied and sparse across the resources except for a few key comprehensive documents. Resources cover many different factors around supporting nutrition for children with different disabilities and clinical manifestations. 21/30 (70%) resources provide recommendations for at least one aspect of clinical care but only 9/30 (30%) cover at least half of the topics. Recommendations for managing feeding are largely absent from the guidelines. UNICEF's guidance on Including Children with Disabilities in Humanitarian Action: Nutrition is an exception and demonstrates how disability-specific recommendations can be integrated into guidelines. The most comprehensive resources are from Holt International, LEARN, Ubuntu and MAITS, which were intentionally designed to improve skills related to feeding and nutrition to support children with disabilities. The most common recommendations were around positioning, timing, frequency and duration of feeding, feeding environment and hygiene, and feeding equipment each found in 12 different resources.

> *"The risk of aspiration, and chronic upper respiratory infections … These things can start pushing children into that cyclical interaction with malnutrition, disability and infection." Participant 5*

There are also some topics and populations not covered as comprehensively in the resources, including nutrition quality of diets, sensory challenges in feeding, hydration, reflux, vomiting and wider medical support. KIs highlighted that the quality of the diet is a priority area for all nutrition programming but has particular importance for children on modified diets, and children whose diet may be very restricted due to food aversion and sensory sensitivities to food. Resources on the modification of food or managing restricted eating rarely include consideration of the nutritional quality of the diet.

> *"We find children with disabilities end up being bottle fed for a long time because they think this is the only thing they're used to, it's the only thing they'll tolerate. They're getting rice porridge in a bottle for the next 10 years, which is not nutritionally adequate." Participant 5*

Key informants discuss how recommendations for a nutritionally balanced and modified diet will look different in different contexts depending on access to and acceptability of varying ingredients and cooking methods, particularly for thickening foods and recommendations need to consider the quality and diversity of macro- and micro-nutrients.

> *"Up to 65% of children generally in [place redacted] are anaemic, so you wonder what percentage of children with disabilities, who have an even worse diet, would be, and how much does that restrict their improvements." Participant 2*

**Table 4. Heat map demonstrating key content areas under the theme of Management of Feeding and Malnutrition, in the resources from the Feeding and Disability Resource Bank. [22] Red indicates not acknowledged; light blue: acknowledged but no guidance; medium blue: acknowledged and some guidance (1-2 recommendations); and dark blue: acknowledged and comprehensive guidance (>3 recommendations).**

| Resource | Breastfeeding support | Positioning | Feeding Equipment | Feeding Environment and Hygiene | Timing, Frequency and Duration of Feeding | Nutrition Quality and Micronutrients | Food textures / Liquid Consistency | Sensory Aversion / Restricted Food Intake | Dysphagia, Choking and Aspiration | Oral Support, Bite Reflex and Tongue Thrust | Reflux, Vomiting | Nasogastric or Oro-gastric Feeding | Hydration | Dental / Oral Health | Responsive / Self Feeding |
|---|---|---|---|---|---|---|---|---|---|---|---|---|---|---|---|
| Feeding in Children with Cerebral Palsy [25] | light blue | dark blue | dark blue | light blue | light blue | light blue | dark blue | red | dark blue | dark blue | red | dark blue | red | red | red |
| Resource library [26] | medium blue | dark blue | light blue | medium blue | light blue | medium blue | red | light blue | red | red | red | red | red | red | red |
| Introduction to Nutrition and Feeding for Children [27] | dark blue | red | red | red | red | medium blue | dark blue | red | red | red | dark blue | dark blue | red | red | light blue |
| Nutrition Integration for Children with Special Needs [28] | dark blue | dark blue | dark blue | medium blue | dark blue | dark blue | dark blue | medium blue | dark blue | dark blue | medium blue | red | light blue | light blue | dark blue |
| Feeding and Positioning Manual: Guidelines for Working with Babies and Children [29] | dark blue | dark blue | dark blue | dark blue | dark blue | red | dark blue | dark blue | dark blue | dark blue | dark blue | red | light blue | dark blue | dark blue |
| Working with Infants with Feeding Difficulties [30] | dark blue | dark blue | dark blue | red | dark blue | red | dark blue | red | dark blue | dark blue | dark blue | dark blue | red | red | red |
| Working with Children with Eating and Drinking Difficulties [31] | red | dark blue | dark blue | medium blue | dark blue | light blue | red | dark blue | dark blue | dark blue | dark blue | light blue | dark blue | dark blue | dark blue |
| Our Videos: Care of Small Babies [32] | dark blue | dark blue | medium blue | red | medium blue | red | dark blue | dark blue | red | red | red | dark blue | red | red | red |
| How to approach Feeding Difficulties in young Children [33] | red | red | red | light blue | light blue | red | red | dark blue | red | red | medium blue | red | red | red | light blue |

*(Continued)*

**Table 4.** (Continued)

| | Breast-feeding support | Posi-tion-ing | Feeding Equip-ment | Feeding Environ-ment and Hygiene | Timing, Fre-quency and Duration of Feeding | Nutrition Quality and Micronu-trients | Food textures / Liquid Consis-tency | Sensory Aversion/ Restricted Food Intake | Dysphagia, Chok-ing and Aspiration | Oral Sup-port, Bite Reflex and Tongue Thrust | Reflux, Vomiting | Naso-gastric or Oro-gastric Feeding | Hydra-tion | Dental / Oral Health | Respon-sive / Self Feeding |
|---|---|---|---|---|---|---|---|---|---|---|---|---|---|---|---|
| Assessment of Neurodis-ability and Malnutrition in Children in Africa [34] | | | | | | | | | | | | | | | |
| Growth Charts for Children with Special Health Care Needs [35] | | | | | | | | | | | | | | | |
| Assessment of Growth and Nutrition in Children with Cerebral Palsy [36] | | | | | | | | | | | | | | | |
| Feeding Problems in Infancy and Early Child-hood [37] | | | | | | | | | | | | | | | |
| Responsive Care and Early Learning Addendum [38] | | | | | | | | | | | | | | | |
| Child Nutrition Program: Community Flipbook [39] | | | | | | | | | | | | | | | |
| Ubuntu Man-ual [40] | | | | | | | | | | | | | | | |
| Baby Ubuntu [41] | | | | | | | | | | | | | | | |
| Juntos [42] | | | | | | | | | | | | | | | |
| Working with Children with Developmen-tal Disabilities and Caregiv-ers [43] | | | | | | | | | | | | | | | |

*(Continued)*

**Table 4.** (Continued)

| | Breast-feeding support | Posi-tion-ing | Feeding Equip-ment | Feeding Environ-ment and Hygiene | Timing, Fre-quency and Duration of Feeding | Nutrition Quality and Micronu-trients | Food textures / Liquid Consis-tency | Sensory Aversion/ Restricted Food Intake | Dysphagia, Chok-ing and Aspiration | Oral Sup-port, Bite Reflex and Tongue Thrust | Reflux, Vomiting | Naso-gastric or Oro-gastric Feeding | Hydra-tion | Dental / Oral Health | Respon-sive / Self Feeding |
|---|---|---|---|---|---|---|---|---|---|---|---|---|---|---|---|
| Supporting Families for Nurturing Care [44] | red | red | red | dark blue | medium blue | red | medium blue | light blue | red | red | red | red | red | red | dark blue |
| Prevention Toolkit on Cerebral Palsy [45] | red | light blue | light blue | red | red | light blue | light blue | red | light blue | light blue | red | red | red | red | light blue |
| Seen, Counted, Included [5] | red | red | red | red | red | red | red | red | red | red | red | red | red | red | red |
| Essential Actions on Disability Inclusive Nutrition [46] | red | red | red | red | red | light blue | light blue | red | red | red | red | red | red | red | red |
| Inclusion of Children and Young People with Disabili-ties in Routine General Healthcare [47] | red | red | red | red | red | red | red | red | red | red | red | red | red | red | red |
| Improving Young Chil-dren's Diets during Com-plementary Feeding [48] | red | light blue | light blue | red | medium blue | dark blue | light blue | red | red | red | red | red | red | red | dark blue |
| Disability Inclusive Health Ser-vices Toolkit [49] | red | red | red | red | red | red | red | red | red | red | red | red | red | red | red |
| Monitoring Children's Development in Primary Care Services [50] | red | red | red | red | red | red | red | red | red | red | red | red | red | red | red |

*(Continued)*

Table 4. (Continued)

| | Breast-feeding support | Posi-tion-ing | Feeding Equip-ment | Feeding Environ-ment and Hygiene | Timing, Fre-quency and Duration of Feeding | Nutrition Quality and Micronu-trients | Food textures / Liquid Consis-tency | Sensory Aversion/ Restricted Food Intake | Dysphagia, Chok-ing and Aspiration | Oral Sup-port, Bite Reflex and Tongue Thrust | Reflux, Vomiting | Naso-gastric or Oro-gastric Feeding | Hydra-tion | Dental / Oral Health | Respon-sive / Self Feeding |
|---|---|---|---|---|---|---|---|---|---|---|---|---|---|---|---|
| Nurturing Care Hand-book, Stra-tegic Action 4 [51] | | | | | | | | | | | | | | | |
| Including Children with Disabilities in Humanitarian Action: Nutri-tion [52] | | | | | | | | | | | | | | | |
| A Toolkit for Measuring Early Child Development in LMICs [53] | | | | | | | | | | | | | | | |

The MAITS training programme on developmental disabilities for non-specialists gives guidance on nutrition quality and even contains recipe cards for highly nutritious meals of different textures and consistencies. [43] There is increasing recognition of developmental difficulties and sensory challenges with food, textures and taste. KIs highlight the resources need to also recognize these issues with clinical recommendations for identification and management. Guidance for newborns and young infants with disabilities who are breastfeeding or bottle feeding is also limited. More medical topics, including reflux and vomiting (8/30 resources, 27%), oral support and reflexes (6/30, 20%), dental and oral health (6/30, 20%) and hydration (3/30, 10%) are limited in the resources. Hydration is another key area highlighted as important by KIs but is poorly covered in the resources with only one giving recommendations to consider and monitor hydration status. There are also limited resources on reflux and vomiting for children with low muscle tone or poor positioning. KIs mention how clinicians providing direct services are often nervous about implementing feeding practices.

## Integration of Care

In Table 5, 19/30 (63%) of the resources identify the need for a multi-disciplinary approach to care for children with disabilities but few contain examples of what exactly to include. Examples of holistic care found in the thematic analysis include the emotional and cognitive development of the child, social and psychosocial support for the family and caregiver, and broader recommendations to address barriers in the community such as accessibility to services, stigma and discrimination. Recommendations for at least one of these aspects are given in 4/13 (31%) training manuals, 6/8 (75%) programme packages and 8/9 (89%) guidelines. Only 9/30 (30%) resources include recommendations to address more than three aspects of wider care. The Nurturing Care Framework, the Ubuntu Programmes and UNICEF's Including Children with Disabilities in Humanitarian Action include recommendations for six or seven different aspects of holistic support. [40–42,44,52]

## Support for the child

Key informants report children with disabilities are less likely to access other health services or be reached by programmes in places such as schools. These children are more likely to have comorbidities, increasing their risk of malnutrition, and poor response to treatment and support. One describes the challenges children with malnutrition experience with treatment and rehabilitation in physiotherapy.

> *"A lot of the referrals that we were getting were children who needed corrective and acute management of malnutrition before we could start any rehabilitation process because any energy they were using we had to preserve for survival." Participant 2*

Some of the programmes connect feeding with support for the child's cognitive, social and emotional development. However, these recommendations are less common in the training resources and programmatic guidance and therefore KIs identify that unless those specific programmes are being implemented, the child's socio-emotional and cognitive development goes unconsidered.

## Support for the family/caregivers

A common theme from interviews is the impact of disability on the child and their family or caregiver. However, resources with information for practitioners to support caregivers are limited. Only one of the training resources (8%) and two programme guidelines (22%) give any recommendations to support the families' needs, which is incongruent with KI descriptions of the necessity of additional support such as kitchen gardens, cooking demonstrations and finance programmes.

**Table 5. Heat map demonstrating key topics related to integration of care. Red indicates not acknowledged; light blue: acknowledged but no guidance; medium blue: acknowledged and some guidance (1-2 recommendations); and dark blue: acknowledged and comprehensive guidance (>3 recommendations).**

| | Multidisciplinary Management and Referrals | Social and Emotional Development | Play, Learning and Cognitive Development | Maternal and Family Psychosocial Wellbeing | Supporting the Family | Accessibility | Awareness and Rights | Addressing Stigma and Discrimination |
|---|---|---|---|---|---|---|---|---|
| Feeding in Children with Cerebral Palsy [25] | medium blue | light blue | light blue | red | red | red | red | red |
| Resource Library [26] | red | red | red | red | light blue | red | light blue | red |
| Introduction to Nutrition and Feeding for Children [27] | red | red | red | red | red | red | red | red |
| Nutrition Integration for Children with Special Needs [28] | light blue | dark blue | red | red | red | light blue | red | red |
| Feeding and Positioning Manual: Guidelines for Working with Babies and Children [29] | dark blue | dark blue | dark blue | dark blue | red | red | red | red |
| Working with Infants with Feeding Difficulties [30] | red | red | red | red | red | red | red | red |
| Working with Children with Eating and Drinking Difficulties [31] | dark blue | light blue | light blue | light blue | dark blue | red | red | red |
| Our Videos: Care of Small Babies [32] | medium blue | medium blue | red | red | red | red | red | red |
| How to Approach Feeding Difficulties in Young Children [33] | red | light blue | red | red | red | red | red | red |
| Assessment of Neurodisability and Malnutrition in Children in Africa [34] | red | light blue | light blue | red | light blue | light blue | light blue | light blue |
| Growth Charts for Children with Special Health Care Needs [35] | red | red | red | red | red | red | red | red |
| Assessment of Growth and Nutrition in Children with Cerebral Palsy [36] | red | red | red | red | red | red | red | red |
| Feeding Problems in Infancy and Early Childhood [37] | dark blue | light blue | red | red | light blue | red | red | red |
| Responsive Care and Early Learning Addendum [38] | medium blue | dark blue | dark blue | dark blue | red | medium blue | red | medium blue |
| Child Nutrition Program: Community Flipbook [39] | red | medium blue | red | red | red | red | red | red |
| Ubuntu Manual [40] | dark blue | dark blue | dark blue | light blue | dark blue | dark blue | dark blue | dark blue |
| Baby Ubuntu [41] | dark blue | dark blue | dark blue | light blue | dark blue | dark blue | dark blue | dark blue |
| Juntos [42] | dark blue | dark blue | dark blue | light blue | medium blue | dark blue | dark blue | dark blue |
| Working with Children with Developmental Disabilities and Caregivers [43] | medium blue | dark blue | dark blue | light blue | dark blue | red | red | red |
| Supporting Families for Nurturing Care [44] | light blue | dark blue | dark blue | dark blue | dark blue | dark blue | dark blue | dark blue |
| Prevention Toolkit on Cerebral Palsy [45] | red | light blue | light blue | red | light blue | red | red | red |

*(Continued)*

| | Multi-disciplinary Management and Referrals | Social and Emotional Development | Play, Learning and Cognitive Development | Maternal and Family Psychosocial Wellbeing | Supporting the Family | Accessibility | Awareness and Rights | Addressing Stigma and Discrimination |
|---|---|---|---|---|---|---|---|---|
| Seen, Counted, Included [5] | medium blue | red | medium blue | light | medium blue | medium blue | dark blue | medium blue |
| Essential Action on Disability Inclusive Nutrition [46] | dark blue | red | medium blue | red | light | dark blue | medium blue | medium blue |
| Inclusion on Children and Young People with Disabilities in Routine General Healthcare [47] | dark blue | red | light | light | red | light | dark blue | dark blue |
| Improving Young Children's Diets during Complementary Feeding [48] | light | red | red | red | red | red | red | light |
| Disability Inclusive Health Services Toolkit [49] | dark blue | red | red | red | red | light | dark blue | dark blue |
| Monitoring Children's Development in Primary Care Services [50] | medium blue | red | red | medium blue | light | dark blue | dark blue | light |
| Nurturing Care Handbook, Strategic Action 4 [51] | red | red | red | red | red | light | medium blue | red |
| Including Children with Disabilities in Humanitarian Action: Nutrition [52] | medium blue | red | red | red | medium blue | dark blue | dark blue | dark blue |
| A Toolkit for Measuring Early Child Development in LMICs [53] | red | light | light | light | light | red | red | red |

*"Most of them come from socially disadvantaged backgrounds, the risk is heightened for children with disabilities. If we look at the mothers, they are also malnourished and they are also at a higher risk of giving birth to a child with a cleft lip and palate." Participant 1*

KIs felt this relationship with poverty is not addressed adequately in programmes, training or guidelines.

*"Broader support to the family is so important. Thinking about if this child comes to me, do I have a way to check that they are at least connected to someone who can help them navigate whether or not there's social protection programmes they can enroll into." Participant 10*

Three resources (10%) give recommendations for a caregiver's emotional and psychosocial health or wellbeing. There are also three other resources on caregiver psychosocial support in the resource bank, but these are excluded from this analysis due to a lack of any reference to the relationship to disability or nutrition.

## Context

Two resources are developed for children in specific contexts: emergency and humanitarian needs. KIs indicate the overarching guidance in many resources are useful in a wide variety of settings, but they also speak of the number of contextual gaps missing from the literature and the value of contextually adapted recommendations for improving the specific care and support needed for individual children.

*"The institutional care setting is often overlooked [and many children with disabilities live in care], and it is such a prime example of nutrition affecting disability, disability affecting nutrition, all affecting caregiving. It's the whole cycle of it all. Resources for that context are severely lacking." Participant 7*

Only 2/30 (6%) resources mention children in institutional care.

## Stigma and discrimination

Key informants highlight the barriers to children accessing appropriate nutrition support as a key issue. Five of the resources discuss wider environmental and community factors preventing children from accessing care, including stigma and discrimination. Recommendations or resources for addressing societal stigma and supporting children's rights are limited in training manuals.

*"We have to consider that in a lot of these communities, there's a ton of stigma around disability and misperceptions of what causes disability. A lot of our health workers come from those same communities. A lot of times, the health workers hold those same perceptions and blame mothers." Participant 10*

*"How much is because of the feeding difficulty related to their disability, and how much of it is stigma, how much of it is poverty, what is the root cause of these children's malnutrition? Our gut feeling is it's about 30% feeding difficulties related to the disability and the rest is more poverty or stigma." Participant 2*

## Recommendations on future development of resources

KIs identified several mechanistic pathways linking the inadequate resource base to poor outcomes in care received by children with disabilities.

1. Identification – Weak guidance on identifying feeding difficulties or disabilities means children are not screened, admitted or monitored correctly resulting in their exclusion from programmes, their health needs not being met and persistent malnutrition.

2. Management/Resource Ecosystem – Fragmented recommendations, guidance and resources can lead to persistent gaps, suboptimal training, inconsistent nutritional practice and poor implementation of programming. Key-informants identify multiple gaps in the available guidance including recommendations for appropriate screening and assessment tools, nutritional quality of modified diets, and comorbid health conditions such as micronutrient deficiencies, dehydration and reflux.

3. Integration of Care/ Holistic Environment – Stigma and poverty, combined with strain on caregivers, can impact mealtimes and feeding practices resulting in malnutrition. That same stigma and discrimination can result in reduced presentation or engagement with nutrition services, worsening malnutrition outcomes. Without addressing a child's integrated needs, nutrition will continue to be a challenge Fig 2.

To improve this, recommendations from KIs focus on the need for standardized resources and recommendations across the three areas:

- identification of disabilities, feeding difficulties and malnutrition,

- management of disabilities and malnutrition, and

- integration of wider care for children with disabilities.

| Gaps | Standardise | Operationalise |
|---|---|---|
| **Identification**<br>• Screening tool to monitor development and disability in younger infants 0-2 years<br>• Recommendations for growth monitoring and malnutrition<br>• Formal method of feeding assessment | • Which development tools are most relevant to the nutrition context<br>• Caveats for inclusion criteria when anthropometric indicators are suitable | • How developmental monitoring, disability screening and identification of feeding difficulties should be implemented in nutrition programming |
| **Management**<br>• Guidance on nutrition quality in the context of restricted or modified diets<br>• Recommendations for managing hydration, reflux, vomiting, and more medically severe feeding difficulties | • Include and make more comprehensive recommendations for management into more program packages and programmatic guidance<br>• Inclusion of social, emotional and cognitive development and how this relates to feeding | • Include references to more comprehensive training manuals in guidelines |
| **Integration**<br>• Rights of a child with a disability, and approaches to the challenges of accessibility, stigma and discrimination in training materials (specific to nutrition context)<br>• Recommendations for vulnerable populations, such as those in institutional care, refugees, and migrants | • Guidance on social, financial and emotional support for families of children with disabilities<br>• Caveats to accommodate a child's need for longer-term care and support in nutrition programming | • Programmatic guidance and packages on how to implement an integrated and holistic clinic for children with disability<br>• Provide implementation guidance on how to identify and establish referral links between care |

**Fig 2. Suggestions from Key-Informants on how to strengthen the Rresources.**

Within each of these three sections, gaps in resources, the need for standardized recommendations and guidance on how to operationalize are highlighted as key points of issue. KIs describe the challenge for direct service providers in using the many currently existing resources to achieve a comprehensive needs assessment and management strategy, highlighting the need to use several resources. A common theme includes standardising the existing recommendations and resources. For example, having comprehensive manuals for disability-specific nutrition training and simplified recommendations for care of individuals with disabilities integrated into broader guidelines. KIs suggest the most important topics to include are tools for screening development and disability, red flags for feeding difficulties (during breastfeeding, bottle-feeding and after transitioning to solid foods) and first-line management strategies of positioning and texture and consistency modification for a nutritious diet. Commonly suggested recommendations include tools to direct professionals to specific resources or tools they need. Integration of disability services into existing programming is also identified as a key priority area, especially for breastfeeding programming or complementary feeding programming which are also necessary for children with disabilities. KIs discuss the need for guidance on how to improve existing nutrition programmes to be disability-inclusive or when it is best to have standalone disability programmes or how to integrate disability recommendations into programming.

*"We are missing more operational guidance. What does it look like to be inclusive? What is it that we really want to see happen in those programmes? We hear it from our country offices 'what is it exactly that you're asking us to do?'" Participant 10*

One example given suggests recommendations for multi-disciplinary care to be accompanied by guidance on what exactly to include, in what settings and how to develop the expertise or referral pathways so practitioners know what to expect when implementing the programmes. KIs acknowledge that hesitation to establish formal guidance is due to weak evidence on these issues.

*"We don't have great data informing, 'If you do this, you're going to have these results,' but we certainly know that if you do nothing, you're going to have these really bad results. We know some things we can be doing and we need to understand how to do them better." Participant 10*

The more disability is integrated into nutrition programming, the more opportunity it provides for implementation research. Some programme packages we reviewed were particularly strong, especially those by Ubuntu. Thus, instead of establishing entirely new programmes, implementation research could help refine and standardize existing ones.

*"More implementation research to understand those pieces so that the tools themselves can be strengthened and streamlined to what really works best." Participant 10*

Based on coverage of key content and depth of recommendations in the heat maps, these resources are identified as particularly strong and suitable for priority use in the short-term:

• MAITS and Holt's manuals for training professionals on disability-specific feeding and nutrition

• Ubuntu programmes for educating caregivers on appropriate feeding and development

• UNICEF's guideline, Including Children with Disabilities in Humanitarian Action: Nutrition, for strengthening disability inclusion in mainstream nutrition programming.

**Identified barriers and facilitators to implementation**

KIs highlight that there are external factors both promoting and inhibiting uptake and implementation of current guidelines. These barriers and facilitators include the visibility of disability in nutrition agendas, access to the relevant resources and the capacity of the system to adopt the recommendations. These priority topics must be addressed for improved future practice (Fig 3).

**Accessibility**

Accessibility is identified by KIs as a key factor influencing implementation of the resources, including physical accessibility to the resource literature (i.e., printed copies, internet access), language accessibility (i.e., translating manuals into more languages/ translation costs), and contextualisation of resources for relevance to local contexts, languages and geographies. Not having a central location for resources and having to search broadly (i.e., newsletter, journals, webinars, websites) is identified as a key barrier to accessibility. The Feeding and Disability Bank is highlighted as addressing this problem well, providing a centralised platform for resources to be more accessible. Even then, however, access to some of the Feeding and Disability Bank resources were limited (i.e., only accessible by entering personal details, through organisations' own internal pages or behind paywalls). Eight resources from the bank were excluded from this thematic analysis because they were not open access. KIs reflect that even though electronic access is improving, there remains accessibility issues for many countries, limiting dissemination and utilization of resources. Many of the larger programme packages and training manuals are now available in different languages, but translated versions are limited, mainly due to translation costs (S1 Table). Existing translations may also be poor or lack contextualization.

*"The interpreters that we use at [organisation redacted] for all of our training and for the actual translation of the manual, have been outstanding. They will even give us feedback in real time like, 'That's not going to make sense to them or maybe we should try to say in a different way, because that might come across differently.'" Participant 12*

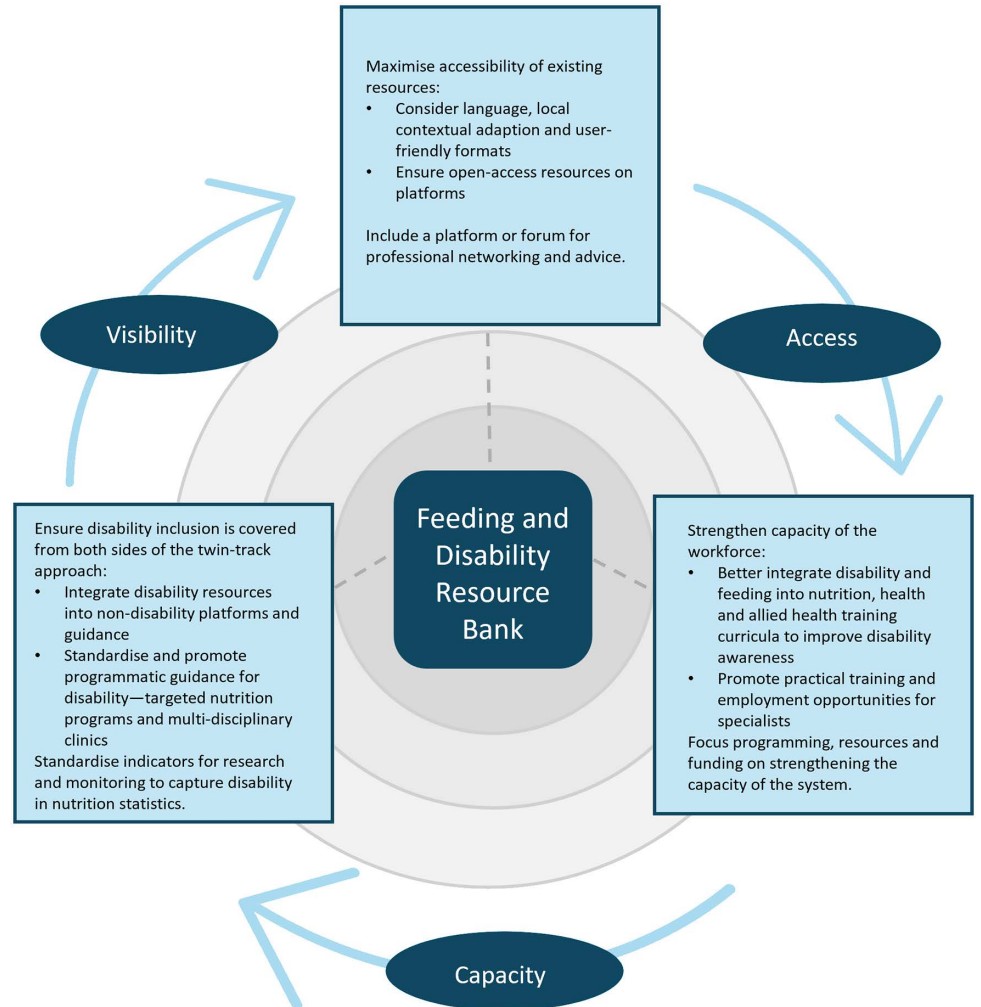

**Fig 3. Recommendations for removing barriers to the implementation of disability and nutrition resources across the themes of visibility, access and capacity.**

The applicability of recommendations within the local context is also important. One KI describes how developmental milestones for feeding skills can differ depending on the child's environment and prevailing cultural norms. Many training manuals and programmes also now include recommendations and guidance on local adaption (S1 Table). Also, the strength of personal and professional networks for sharing resources was a strong theme throughout the interviews. Informal communication networks, such as WhatsApp are described as one method for sharing existing resources, advice and support. First-hand experiences from professional networks are also mentioned as particularly helpful:

> *"Using other people's experiences on the ground, 'How does this work in this setting?', is definitely more helpful than guidelines that then have to be adapted to disability, or to this setting." Participant 2*

Several KIs mention using online discussion platforms, working groups or listservs as useful methods for resource sharing. These support building networks and sharing case-studies and advice, advocacy and identification of future priorities for the field.

## Capacity

Despite resources being visible and accessible on the bank, barriers are also identified in workforce capacity, understanding and ability to implement recommendations within routine clinical practice.

*"There's a big gap in practical training for staff. Time and time again, that's the feedback we've had that they just don't know how to apply it." Participant 8*

Many of the resources include training programmes for health workers, or have supplementary pictures, videos and even participatory learning activities to help improve understanding of the recommendations (S1 Table). However, more focus on disability/nutrition interactions are needed right from basic/undergraduate training through to post-qualification/post-graduate training. This would ensure that staff have not only theoretical understanding but also the practical knowledge and expertise to use the strategies in everyday clinical practice. Improvements in integrated curricula on care for children with disabilities is discussed as impacting later use of resources, as many current courses do not include any training on disability.

*"Disability is there, like it or not, it's part of us. To make it something you learn after you qualify, that's not good." Participant 8*

Other limitations reported to use of resources include: time, competing demands from other duties, funding and capacity of practitioners/staff:

*"It's really important to think about who we are addressing when we are producing these manuals. The same nurses who have to do the bloods, the weights, are now expected to also help with the feeding. It can't become a part of people's existing job because it just doesn't get done." Participant 8*

Multi-track approaches are considered essential but hard to manage for staff with limited time or knowledge, such as providing direct feeding therapy for children and providing feeding training for caregivers. Some of the resources are written so that all individuals, regardless of clinical background, could implement the practices.

## Visibility

Both the thematic analysis and interviews show international commitment to disability is growing in the field of nutrition, especially with the creation of guidance from UNICEF's Disability Inclusion Policy and Strategy. [63] However, interviews also reveal one of the limitations of a resource bank; that it only targets one side of a twin-track approach to disability inclusion; tailoring disability-specific programmes, without also ensuring integration into mainstream programming and guidance.

*"Where it is now, it is those who are interested, and those who are struggling with this group of children, who are looking for these resources and trying to apply them, but it's not systemic." Participant 10*

*"[The International Milk Code], that's a big one, talking about breastfeeding and breast milk substitutes and there's no mention of children with disabilities and what their needs might be." Participant 5*

KIs also discuss the lack of data collection for children with disabilities, limiting inclusion and development of resources or evaluation of programmes/practices. The resource, "Seen, Counted, Included" highlights the exclusion of children with disability from research and data surveillance, as one in four children with disability had missing height measurements

and one in five missing their weight measurements, compared to 1 in 20 and 1 in 30 respectively for children without disability. [5] KIs raise the need for more nuanced indicators for children with disability to reduce the risk of those children being excluded from programmes and to capture the impact of treatment on functioning and quality of life (i.e., quality of life indicators may improve far more than anthropometric indicators).

*"Nutritional support might be changing quality of life and ease of care." Participant 4*

## Discussion

This thematic analysis demonstrates the global resource-base for disability-focused nutrition guidance is incomplete, scattered, and sometimes even conflicting in recommendations. Alongside this emerges several mechanistic pathways for how poor resources, guidance and programming are in turn impacting children's outcomes including suboptimal identification of children at risk, inadequate professional knowledge on how to manage malnutrition and feeding difficulties, and missed opportunities for a holistic approach to care. Consequently, children with disabilities and their nutritional status can be impacted through a combination of biological challenges, fragmented incomplete guidance, weak health systems and social limitations. These factors are perpetuated by each other and create cycles that keep children at high risk. The needs of those with disabilities must be considered at all levels of programming and services.

Additionally, this analysis identifies barriers to the uptake, implementation and utilization of resources and inclusive nutrition programming for individuals with disabilities. Without addressing the visibility of disability in nutrition agendas, the accessibility of quality resources and the strength/capacity of systems to enact the recommendations, programming will continue to be suboptimal and exclude children with disabilities.

The paucity of recommendations on disability in nutrition has been documented in other work. [16,20,64] A guideline review of more than 70 international, national and local nutrition protocols found only three with a section dedicated to disability. [65] Health workers can thus be left ill-equipped to manage these challenges and children with disability risk being systematically excluded from or poorly managed in nutrition programmes.

Interviews done at the time of establishing the USAID Feeding and Disability Resource Bank [22] found similar gaps in the resources to those of this research — recommendations for active screening of disability in malnutrition treatment programmes, the need for guidance on providing multidisciplinary management for a child and family, and alternative indicators that can be integrated into nutrition coverage surveys, monitoring and evaluation. In particular, the call for standardised disability screening/assessment tools in nutrition guidance has been documented in other literature. Research demonstrates having disability or feeding screening and assessment for younger infants, such as universal newborn screening, can increase the identification of congenital disorders and delays, thereby improving early intervention and preventing poor health and nutrition outcomes. [67] Another example demonstrated the use of Washington Group Questionnaires in a cohort of malnutrition survivors in Malawi identified many physical and behavioral difficulties which previous clinical assessment had missed. [14] The feasibility of integrating developmental and disability screening into existing child health and nutrition programmes has been demonstrated in a systematic review. [66] Our research found there were some higher-quality resources, giving a more comprehensive account of how to identify children at risk of malnutrition and disability, how to appropriately manage their feeding and nutrition, and how to integrate these within a broader approach to care. However, these are currently underutilized.

Our research identifies a lack of access to resources, a lack of visibility of disability in nutrition agendas and inadequate capacity of the global health system to adequately respond, as barriers to implementation.

The USAID interviews identified similar barriers to the implementation of the resource literature including poor knowledge among health and nutrition professionals on identification of feeding difficulties and how they relate to disabilities. [21] This lack of knowledge was found to be driven by inadequate practical training for staff, insufficient funding to improve training and inadequate standardized policies, programmes and evidence to follow.

This research and others, speak to the need for integration of recommendations into global guidance and policy, considering implementation in both disability-specific and inclusive nutrition programming. Shifts in global funding make this research important as policy makers develop new strategies to create inclusive health care systems for individuals with disabilities.

## Strengths and limitations of the study

This project builds on previous work done in compiling the resource bank, offering detailed information on exactly which topics and issues are covered in the resources, including neglected topics such as hydration. It also offers valuable user perspectives on those resources via our KI interviews. We acknowledge other relevant resources may exist outside of the resource bank (such as those in local languages), resources added and created since we did this review or those on the resource bank may have had minor changes, as the literature is constantly evolving. We were also unable to review some resources due to them not being open source (e.g. SPOON's feeding and nutrition package for vulnerable children, highlighted as useful by one of our KIs). Mixed methods improved the rigour and value of our study by placing results of the resource analysis in the context of their implementation. We took steps to avoid bias by having a wide KI interview base and conducting interviews to a point of saturation. We do, however, acknowledge limitations due to small sample size, relying on snowball sampling and having only one coder. The key-informants chosen varied in background but were English-speaking and mid- to full-career and may not represent all perspectives of individuals working in the field of nutrition and child disability. We believe that additional resources and interviews would not have changed our overall conclusions. However, explicitly addressing the limitations of this paper ensures that reviewers are fully informed and provides areas for future research to address.

## Recommendations

Several key recommendations to improve disability inclusion arose during this research with implications for programming, resource development and policy. One priority research area is implementation research of existing resources and programmes to help refine and adapt recommendations in different contexts. This includes local validation of the assessment and screening tools to improve the identification of children in programmes. Similarly, research is needed on appropriate anthropometric and outcome indicators that could be used to identify and monitor children with disability. Having these indicators would also enable researchers to capture the scope of disability in programmes as an advocacy tool to better direct allocation of resources and funding.

Considering the high care demands on health care systems, communities, caregivers and the need for community capacity-strengthening rising to the forefront in this report, an important follow-up would be a similar analysis of the resources intended for use by the community-specific programmes or by caregivers. Understanding/expanding the adaptability of key community resources, such as MAITS' Guide for Parents: Caring for a Child with Developmental Disabilities and Holt's Community Resource Manual, would strengthen the overall picture of the available literature. [68,69] Current practice should use these results to identify resources most suitable to individual contexts or prioritise high-quality appropriate resources.

Other research could build on this analysis by using an evaluation tool, such as the AGREE checklist, to determine the standard of the existing resources and clinical guidelines. [70]

Although gaps still remain, this project highlights that existing recommendations and management options could already be doing more to improve disability inclusion and support. Integrating these resources into policy could shift the dynamic and enable a more proactive approach to earlier identification of underlying disability and improve care for children. To further ensure equitable opportunities for children with disabilities, policy development should focus on utilising evidence-based resources to create more inclusive programming and services.

## Conclusion

In our thematic analysis of currently available resources, there are few training materials, programme packages or guidelines that provide comprehensive recommendations specific to the nutritional management of children with disability. Several mechanistic pathways link a limited resource base to inadequate quality of care for children with disabilities, including insufficient identification of at-risk children, suboptimal professional knowledge and training for managing malnutrition, and ongoing barriers to efficient holistic care such as stigma and discrimination. Priorities for future development of the global resource-base include addressing neglected topics, standardising recommendations, and operationalising guidance for disability inclusion in both broader nutrition and disability-specific programming. Consideration must also be given to the human, financial and resource capacity of health systems to implement those recommendations, and the visibility and accessibility of the resources globally as barriers to optimal use of the resources.

## Supporting information

**S1 Table. Titles, Publisher and Date of Resources included in the Thematic Analysis with Format, Languages and Disabilities included.**
(DOCX)

**S1 File. Key Informant Interview Topic Guide.**
(DOCX)

## Acknowledgments

We extend our gratitude to Ali Murray for her editorial review and the Key Informants for their valuable contributions.

## Author contributions

**Conceptualization:** Katie Fulford, Marko Kerac.

**Data curation:** Katie Fulford.

**Formal analysis:** Katie Fulford.

**Methodology:** Katie Fulford.

**Supervision:** Emily DeLacey, Marko Kerac.

**Writing – original draft:** Katie Fulford.

**Writing – review & editing:** Emily DeLacey, Fiona Beckerlegge, Julia Hayes, Himali de Silva, Marko Kerac.

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
