## [Decision Letter · Decision Letter 0]

1 Jul 2025

PGPH-D-25-01205

Supporting Children with Disabilities in Nutrition Programmes: thematic analysis of training and guidelines key informant interviews

Dear Dr. Fulford,

Thank you for submitting your manuscript to PLOS Global Public Health. After careful consideration, we feel that it has merit but does not fully meet PLOS Global Public Health’s publication criteria as it currently stands. Therefore, we invite you to submit a revised version of the manuscript that addresses the points raised during the review process.

The reviewers comments are below, and include requests to improve the clarity of reporting of your study. Please ensure all comments are addressed in your revision.

Please note that we have only been able to secure a single reviewer to assess your manuscript. We are issuing a decision on your manuscript at this point to prevent further delays in the evaluation of your manuscript. Please be aware that the editor who handles your revised manuscript might find it necessary to invite additional reviewers to assess this work once the revised manuscript is submitted. However, we will aim to proceed on the basis of this single review if possible.

We look forward to receiving your revised manuscript.

Kind regards,

Jennifer Tucker, PhD

Staff Editor

Journal Requirements:

Additional Editor Comments (if provided):

Reviewers' comments:

Reviewer's Responses to Questions

**Comments to the Author**

1. Does this manuscript meet PLOS Global Public Health’s publication criteria? Is the manuscript technically sound, and do the data support the conclusions? The manuscript must describe methodologically and ethically rigorous research with conclusions that are appropriately drawn based on the data presented.? Is the manuscript technically sound, and do the data support the conclusions? The manuscript must describe methodologically and ethically rigorous research with conclusions that are appropriately drawn based on the data presented.

Reviewer #1: Yes

2. Has the statistical analysis been performed appropriately and rigorously?

Reviewer #1: Yes

3. Have the authors made all data underlying the findings in their manuscript fully available (please refer to the Data Availability Statement at the start of the manuscript PDF file)?

The PLOS Data policy requires authors to make all data underlying the findings described in their manuscript fully available without restriction, with rare exception. The data should be provided as part of the manuscript or its supporting information, or deposited to a public repository. For example, in addition to summary statistics, the data points behind means, medians and variance measures should be available. If there are restrictions on publicly sharing data—e.g. participant privacy or use of data from a third party—those must be specified.requires authors to make all data underlying the findings described in their manuscript fully available without restriction, with rare exception. The data should be provided as part of the manuscript or its supporting information, or deposited to a public repository. For example, in addition to summary statistics, the data points behind means, medians and variance measures should be available. If there are restrictions on publicly sharing data—e.g. participant privacy or use of data from a third party—those must be specified.

Reviewer #1: Yes

4. Is the manuscript presented in an intelligible fashion and written in standard English?

Reviewer #1: Yes

Reviewer #1: Introduction

- Line 149-162: The aim and objectives of the study should be incorporated in the Introduction section.

Material & methods:

- Lines 184-186: Justify why these materials were excluded - a comprehensive review to support a policy should include resources intended for all populations.

- Line 198: Provide specific data on the job specification of KI

- Line 210: Has this guide been pilot-tested for feasibility?

- Line 211: How many main and probing questions are in the guide?

- Line 229: How many team members were involved during the analysis? How do you ensure the validity of the analysis?

Results:

- Line 250: Prior to this statement, provide characteristics of reviewed literature - origin country, intended for which type of disabilities, etc.

- Results for literature analysis and KI interview should be separated. The current arrangement of results is difficult to understand and lacks flow for easy readability.

- The use of a heatmap is appropriate; however, the organization of the information is unclear.

Discussion:

- Should start with a summary of findings.

- Lack of elaboration on the key findings and why each key finding is essential in the construction of policy.

- Incoherent from one paragraph to another.

Conclusion:

- Does not conclude the aim and objective of the study.

Acknowledgement:

- Incomplete information

**Do you want your identity to be public for this peer review?** For information about this choice, including consent withdrawal, please see our Privacy Policy..

Reviewer #1: No

---

## [Decision Letter · Decision Letter 1]

21 Nov 2025

PGPH-D-25-01205R1

Supporting Children with Disabilities in Nutrition Programmes: thematic analysis of training and guidelines key informant interviews

Dear Dr. Fulford,

Thank you for submitting your manuscript to PLOS Global Public Health. After careful consideration, we feel that it has merit but does not fully meet PLOS Global Public Health’s publication criteria as it currently stands. Therefore, we invite you to submit a revised version of the manuscript that addresses the points raised during the review process.

We look forward to receiving your revised manuscript.

Kind regards,

Prof Razak M Gyasi, PhD, PD

Academic Editor

Journal Requirements:

Additional Editor Comments:

Dear Authors,

Thank you for your excellent effort in revising this manuscript. The quality has improved substenatially. However, I would like you to attend to and implement very minor but important comments before we can accept the drfat for publication in PGPH.

1. Please, provide the context of the study - the country or region where the nutrition intervention program took place in the title. Also, the second part of the title ("thematic analysis of training and guidelines key informant interviews") appears incomplete as if something is missing. Pleaase revise this for clarity.

2. Data should also go with a plural verb. Revise and replace "data was or is" with "data were or

are" throughout the manuscript.

3. Please, provide clear mechanistic pathways for all important findings at the discussion section of

the paper.

4. Please do your best to clearly contextualize your findings - your discussion should be amiable to

both localised but should highlight international relevance in order to globalize your study for a wider audience.

5. Explain how the study limitations did not undermine the veracity of the findings.

6. Do NOT include tracked changes in the revised manuscript anymore. Rather clean it all up and provide a

highlight (perhaps in yellow) of all the new changes to the manuscript.

I wish you the best

Razak

Reviewers' comments:

Reviewer's Responses to Questions

**Comments to the Author**

Reviewer #1: All comments have been addressed

publication criteria? Is the manuscript technically sound, and do the data support the conclusions? The manuscript must describe methodologically and ethically rigorous research with conclusions that are appropriately drawn based on the data presented.? Is the manuscript technically sound, and do the data support the conclusions? The manuscript must describe methodologically and ethically rigorous research with conclusions that are appropriately drawn based on the data presented.

Reviewer #1: Yes

3. Has the statistical analysis been performed appropriately and rigorously?

Reviewer #1: Yes

4. Have the authors made all data underlying the findings in their manuscript fully available (please refer to the Data Availability Statement at the start of the manuscript PDF file)?

The PLOS Data policy requires authors to make all data underlying the findings described in their manuscript fully available without restriction, with rare exception. The data should be provided as part of the manuscript or its supporting information, or deposited to a public repository. For example, in addition to summary statistics, the data points behind means, medians and variance measures should be available. If there are restrictions on publicly sharing data—e.g. participant privacy or use of data from a third party—those must be specified.requires authors to make all data underlying the findings described in their manuscript fully available without restriction, with rare exception. The data should be provided as part of the manuscript or its supporting information, or deposited to a public repository. For example, in addition to summary statistics, the data points behind means, medians and variance measures should be available. If there are restrictions on publicly sharing data—e.g. participant privacy or use of data from a third party—those must be specified.

Reviewer #1: Yes

5. Is the manuscript presented in an intelligible fashion and written in standard English?

Reviewer #1: Yes

Reviewer #1: All previous comments and suggestion have been addressed appropriately. Well done.

**Do you want your identity to be public for this peer review?** For information about this choice, including consent withdrawal, please see our Privacy Policy..

Reviewer #1: **Yes:**Nurul Hazirah Binti JaafarNurul Hazirah Binti JaafarNurul Hazirah Binti JaafarNurul Hazirah Binti Jaafar

---

## [Decision Letter · Decision Letter 2]

11 Mar 2026

Children with Disabilities in Nutrition Programmes: thematic analysis of training and guidelines

PGPH-D-25-01205R2

Dear Miss Fulford,

We are pleased to inform you that your manuscript 'Children with Disabilities in Nutrition Programmes: thematic analysis of training and guidelines' has been provisionally accepted for publication in PLOS Global Public Health.

Best regards,

Prof Razak Gyasi, PhD, PD

Academic Editor

Reviewer Comments (if any, and for reference):

Reviewer's Responses to Questions

**Comments to the Author**

Reviewer #1: All comments have been addressed

publication criteria? Is the manuscript technically sound, and do the data support the conclusions? The manuscript must describe methodologically and ethically rigorous research with conclusions that are appropriately drawn based on the data presented.? Is the manuscript technically sound, and do the data support the conclusions? The manuscript must describe methodologically and ethically rigorous research with conclusions that are appropriately drawn based on the data presented.

Reviewer #1: Yes

3. Has the statistical analysis been performed appropriately and rigorously?

Reviewer #1: Yes

4. Have the authors made all data underlying the findings in their manuscript fully available (please refer to the Data Availability Statement at the start of the manuscript PDF file)?

The PLOS Data policy requires authors to make all data underlying the findings described in their manuscript fully available without restriction, with rare exception. The data should be provided as part of the manuscript or its supporting information, or deposited to a public repository. For example, in addition to summary statistics, the data points behind means, medians and variance measures should be available. If there are restrictions on publicly sharing data—e.g. participant privacy or use of data from a third party—those must be specified.requires authors to make all data underlying the findings described in their manuscript fully available without restriction, with rare exception. The data should be provided as part of the manuscript or its supporting information, or deposited to a public repository. For example, in addition to summary statistics, the data points behind means, medians and variance measures should be available. If there are restrictions on publicly sharing data—e.g. participant privacy or use of data from a third party—those must be specified.

Reviewer #1: Yes

5. Is the manuscript presented in an intelligible fashion and written in standard English?

Reviewer #1: Yes

Reviewer #1: The manuscript has been restructured to align with rigorous scientific standards, featuring a good methodology, transparent results, and a well-justified discussion. These findings contribute significantly to bridging the existing knowledge gap in care coordination for children with disabilities.

**Do you want your identity to be public for this peer review?** For information about this choice, including consent withdrawal, please see our Privacy Policy..

Reviewer #1: No
